# Global Burden of Vitamin A Deficiency in 204 Countries and Territories from 1990–2019

**DOI:** 10.3390/nu14050950

**Published:** 2022-02-23

**Authors:** Tian Zhao, Shiwei Liu, Ruijie Zhang, Zhenping Zhao, Hu Yu, Liyuan Pu, Li Wang, Liyuan Han

**Affiliations:** 1Key Laboratory of Diagnosis and Treatment of Digestive System Tumors of Zhejiang Province, Hwa Mei Hospital, University of Chinese Academy of Sciences, Ningbo 315000, China; 2018zhaotian@sina.com (T.Z.); zhangruijie111@163.com (R.Z.); yhfoster@163.com (H.Y.); dannie18@163.com (L.P.); 2Department of Global Health, Ningbo Institute of Life and Health Industry, University of Chinese Academy of Sciences, Ningbo 315000, China; 3National Center for Chronic and Non-Communicable Disease Control and Prevention, Chinese Center for Disease Control and Prevention, Beijing 100000, China; shiwei-liu@aliyun.com (S.L.); zhaozhenping@ncncd.chinacdc.cn (Z.Z.); 4Department of Medical Statistics and Epidemiology, School of Public Health, Sun Yat-sen University, Guangzhou 510000, China

**Keywords:** vitamin A deficiency, Global Burden of Disease, incidence, disability-adjusted life years, epidemiology

## Abstract

Vitamin A deficiency (VAD) is one of the important public health issues worldwide. However, a detailed understanding of the incidence and disability-adjusted life years (DALYs) due to VAD in recent years is lacking. We aimed to estimate the incidence and DALYs of VAD at global, regional, and national levels in terms of sex, age, and socio-demographic index (SDI). Using data from the 2019 Global Burden of Disease (GBD) study, the estimated annual percentage change (EAPC) was measured to assess trends in the age-standardized incidence and DALY rates from 1990 to 2019. The global age-standardized incidence and DALY rates of VAD decreased with an EAPC of −3.11% (95% confidence interval (CI): −3.24% to −2.94%) and −2.18% (95% CI: −2.38% to −1.93%), respectively. The age-standardized incidence and DALY rates decreased least in low-SDI regions, which had the highest age-standardized incidence and DALY rates of all SDI regions. Sub-Saharan Africa, especially central sub-Saharan Africa, had the highest age-standardized incidence and DALY rates in 2019. At the national level, Somalia and Niger had the highest age-standardized incidence and DALY rates. The age-standardized incidence and DALY rates were higher in males than in females. Younger children, especially those aged < 5 years in low-SDI regions, had a higher VAD burden than other age groups. Although the global burden of VAD has decreased, future work should aim to improve the prevention and treatment strategies for VAD, particularly in children aged < 5 years in countries and territories with low SDI values, such as sub-Saharan Africa.

## 1. Introduction

Vitamin A deficiency (VAD), one of the four major nutritional deficiencies worldwide [1], can cause growth and development deficits in children, loss of vision, and increase the risk of infection [2,3]. Furthermore, VAD is a potential risk factor for cognitive impairment and mental illness [4,5]. Therefore, VAD is regarded as the second largest risk factor for the global disease burden, and the life span of populations with disability due to VAD is a crucial quantitative index of burden of malnutrition [6].

According to a report from the World Health Organization (WHO), 190 million preschool children and 19 million pregnant women were exposed to VAD globally [7]. Although the WHO has made great efforts in reducing the epidemic of VAD, due to major differences in the economy, medical care, food carrier, and availability of other nutrients [8], the burden of VAD is unbalanced among population in different regions and is more prevalent in areas with lower socio-economic levels [9,10]. Therefore, an assessment of VAD burden at the regional and national levels and identification of high-risk groups could help to identify the highly endemic areas and populations, which could in turn beneficial to guide effective decision making and resource allocation [11].

To our best knowledge, there are only a limited number of analyses on VAD burden in specific populations and limited regions based on the previous Global Burden of Disease (GBD) study [12,13]. Moreover, the previous GBD analyses have focused more on loss of vision due to VAD rather than VAD itself [2,14]. Currently, a detailed report on the GBD dataset-based analysis of the global VAD burden is lacking. Additionally, previous studies to assess the epidemic of VAD have mainly centered on investigating the prevalence of this condition. However, the annual incidence rate, which is defined as the frequency of new cases per year, expresses more accurately the epidemiological changes of VAD.

Therefore, we used VAD data from the GBD 2019 database to determine temporal trends in the incidence and disability-adjusted life-years (DALYs) of VAD at the global, regional, and national levels and by sex, age, and socio-demographic index (SDI) to systematically summarize the global burden and health development status of VAD.

## 2. Materials and Methods

### 2.1. Overview and Definitions

The methodology used to assess the GBD is continuously improving, and GBD data are updated annually. The 2019 GBD contains annual data of 369 human diseases and 87 risk factors from 204 countries and territories worldwide [15]. We estimated VAD burden every year from 1990 to 2019 worldwide included in 2019 GBD. The study followed the recommendations of the Guidelines for Accurate and Transparent Health Estimates Reporting and was approved by the Medical Ethics Committee of Hwa Mei Hospital. 

### 2.2. Data Sources

Data on the annual incidence, DALYs, and age-standardized incidence and DALY rates of VAD were derived from the 2019 GBD dataset. The Global Health Data Exchange query tool, which is a data source tool online (http://ghdx.healthdata.org/gbd-results-tool, accessed on 10 January 2022), was used to assess health loss from VAD using available epidemiological data. The SDI, which is a comprehensive index that includes per capita gross domestic product, average years of education, and fertility rate to reflect social development, is an important covariate for evaluating disease burden and health development in a certain region. The SDI divides 204 countries and territories into five SDI (high, high-middle, middle, low-middle, and low) quintiles [15]. DisMod-MR 2.1, a Bayesian meta-regression modeling tool, was applied to calculate the data and to determine the 95% uncertainty intervals (UIs) [15]. The DALYs due to VAD were calculated as the sum of years of life lost due to disability and years of life lived with disability. The flowchart for VAD analysis in GBD is shown in Appendix A.

### 2.3. Statistical Analysis

Assuming that the natural logarithm of the age-standardized rate has a linear relationship with time, the estimated annual percentage changes (EAPCs) in the age-standardized rates of VAD were calculated within a specific time frame using the following formula: Y = α + βX + ε(1)

Y represents ln (age-standardized rate), X is calendar year, and ε is the error term. β describes the positive or negative age-standardized rate trend. Therefore,
EAPC = 100 × (exp (β) − 1)(2)

Additionally, 95% confidence intervals (CIs) were determined by the linear model. When the EAPC and CI lower boundary are positive, the age-standardized rate will show an upward trend. Conversely, when the EAPC and CI upper boundary are negative, the age-standardized rate will show a downward trend. Otherwise, the age-standardized rate will show a stable trend [16]. We also determined the correlations between the EAPC and age-standardized rates and between the EAPC and SDI by using Gaussian process regression and Pearson’s correlation coefficient (ρ).

All statistical analyses were carried out using R software (version 3.5.1, the R foundation for statistical computing, Vienna, Austria). A *p*-value of <0.05 was regarded as statistically significant.

## 3. Results

### 3.1. Analysis of Changes in VAD Incidence

The global incidence of VAD decreased from 877,376,294.55 (95% UI: 840,347,028.01–914,976,465.39) to 489,662,708.61 (95% UI: 469,006,373.61–512,234,291.26), which is a decrease of 44.19% from 1990 to 2019 (Appendix A). The global age-standardized incidence rate showed a consistent downward trend. The EAPC was −3.11% (95% CI: −3.24% to −2.94%), from 17,323.23 per 100,000 (95% UI: 16,526.51–18,138.92) in 1990 to 6955.65 per 100,000 (95% UI: 6645.87–7294.23) in 2019, which is a decrease of 59.85%. The age-standardized incidence of VAD decreased in both sexes (male EAPC: −3.36%, 95% CI: −3.45% to −3.23%; female EAPC: −2.75%, 95% CI: −2.92% to −2.51%). However, the age-standardized incidence of VAD was always higher in males than in females from 1990 to 2019, as demonstrated by the male-to-female ratio of 1.31 (Table 1, Figure 1A, and Appendix A).

In 2019, the male-to-female ratio of VAD incidence peaked in the 45–49-year age group globally, in the 0–5-year age group in high-SDI regions, in the 10–14-year age group in high-middle and middle-SDI regions, and in the 45–49-year age group in low-middle- and low-SDI regions (Appendix A). The incidence of VAD decreased with age globally. The incidence was much higher in children aged < 5 years than in other age groups in all SDI regions (Appendix A).

The age-standardized incidence of VAD was much higher in regions with low SDI values than in regions with other SDI values. In 2019, the highest age-standardized incidence of VAD was detected in the low-SDI group (19,156.10 per 100,000; 95% UI: 18,357.37–19,950.88), followed by the low-middle-SDI group (8877.03 per 100,000; 95% UI: 8202.49–9642.84). The high-SDI group showed the lowest age-standardized incidence (586.69 per 100,000; 95% UI: 550.35–625.85). The age-standardized incidence of VAD reduced in all SDI regions during the study period. However, the least decrease in the age-standardized incidence rate was in low-SDI regions where the EAPC was greatest (−2.58%, 95% CI: −2.81% to −2.28%), followed by high-SDI regions (−2.69%, 95% CI: −2.80% to −2.58%) (Table 1, Figure 1A, and Appendix A).

The EAPC was positively associated with the age-standardized incidence (ρ = 0.328, *p* < 0.001) but not significantly associated with the SDI (ρ = −0.028, *p* = 0.492, Appendix A). Low-SDI regions had the highest proportion of incident cases among people aged < 20 years old, in which little change occurred from 1990 to 2019. In contrast, the proportion of incident cases in this age group declined in other SDI regions (Appendix A). The annual proportions of incidence number decreased among people aged < 20 years, but it increased among older adults aged > 40 years, whereas it remained relatively stable in children and adults in all SDI regions from year to year (Appendix A).

At the regional level, in 2019, the highest age-standardized incidence rate of VAD was in central sub-Saharan Africa (25,905.22 per 100,000; 95% UI: 23,288.44–28,883.03), followed by eastern sub-Saharan Africa (23,500.02 per 100,000; 95% UI: 22,337.24–24,765.11) and western sub-Saharan Africa (15,570.91 per 100,000; 95% UI: 14,825.25–16,315.94), whereas the lowest age-standardized incidence rate was noticed in Australasia (148.85 per 100,000; 95% UI: 131.41–168.77), followed by high-income North America (485.64 per 100,000; 95% UI: 408.06–574.05) and Eastern Europe (530.88 per 100,000; 95% UI: 481.60–587.02). From 1990 to 2019, the age-standardized incidence rate decreased least in Oceania (EAPC: −1.01%, 95% CI: −1.25% to −0.67%), followed by Australasia (EAPC: −1.07%, 95% CI: −1.29% to −0.74%) and central sub-Saharan Africa (EAPC: −1.37%, 95% CI: −1.80% to −0.79%), whereas it decreased most in East Asia (EAPC: −5.61%, 95% CI: −5.83% to −5.41%), followed by Southeast Asia (EAPC: −4.64%, 95% CI: −4.73% to −4.50%) and South Asia (EAPC: −4.53%, 95% CI: −4.79% to −4.19%) (Table 1, Figure 1A, and Appendix A).

At the national level, in 2019, the highest age-standardized incidence rate of VAD was in Somalia (63,640.11 per 100,000; 95% UI: 59,279.85–67,657.56), followed by Niger (43,501.51 per 100,000; 95% UI: 38,967.65–48,083.42) and the Federated States of Micronesia (34,768.53 per 100,000; 95% UI: 30,793.63–38,842.13), whereas the lowest age-standardized incidence rate was in Australia (77.34 per 100,000; 95% UI: 66.04–90.92), followed by France (172.51 per 100,000; 95% UI: 141.66–210.35) and Saudi Arabia (190.04 per 100,000; 95% UI: 156.49–229.04). From 1990 to 2019, the age-standardized incidence rates decreased in all countries except Kiribati (EAPC: −0.14%, 95% CI: −0.33% to 0.09%) and Georgia (EAPC: −0.18%, 95% CI: −0.85% to 0.56%), in which the age-standardized incidence rates remained stable. The age-standardized incidence rate decreased least in Somalia (EAPC: −0.30%, 95% CI: −0.35% to −0.28%) and most in Equatorial Guinea (EAPC: −9.90%, 95% CI: −10.55% to −9.45%), followed by Saudi Arabia (EAPC: −8.89%, 95% CI: −9.73% to −7.77%) and the Maldives (EAPC: −8.07%, 95% CI: −8.58% to −7.59%) (Figure 2A, Appendix A). The global EAPCs in VAD age-standardized incidence rates in 204 countries and territories are shown in Appendix A, and age-standardized incidence rates of VAD in 204 countries and territories in 2019 are shown in Appendix A.

### 3.2. Analysis of Changes in DALYs Due to VAD

From 1990 to 2019, global DALYs due to VAD decreased by 40.15% from 1,967,383.01 (95% UI: 1,362,752.37–2,795,112.54) to 1,177,507.28 (95% UI: 805,056.04–1,636,582.37) (Appendix A). The age-standardized DALY rate showed the same trend, with an EAPC of −2.18% (95% CI: −2.38% to −1.93%), from 31.95 per 100,000 (95% UI: 22.11–45.30) to 16.91 per 100,000 (95% UI: 11.5–23.47), which is a decrease of 47.07%. The age-standardized DALY rate decreased in both sexes (male EAPC: −2.50%, 95% CI: −2.67% to −2.30%; female EAPC: −1.75%, 95% CI: −1.99% to −1.44%). However, the age-standardized DALY rate from 1990 to 2019 was always higher in males than in females, as demonstrated by the male-to-female ratio of 1.17 (Table 2, Figure 1B, and Appendix A).

The male-to-female DALY rate of VAD peaked in the 0–5-year age group globally; in the ≥60-year age group in high- and high-middle-SDI regions; and in the 0–5-year age group in middle-, low-middle-, and low-SDI regions (Appendix A). Globally, the DALY rate decreased with age. The highest DALY rate was observed among young people aged < 14 years, especially children aged < 5 years, globally, in all SDI regions (Appendix A).

The age-standardized DALY rate of VAD was much higher in regions with low SDI values than in regions with other SDI values. In 2019, the highest age-standardized DALY rate of VAD was observed in the low-SDI group (38.02 per 100,000; 95% UI: 26.12–52.78), followed by the low-middle-SDI group (19.32 per 100,000; 95% UI: 12.93–27.45). The high-SDI group showed the lowest age-standardized DALY rate (0.29 per 100,000; 95% UI: 0.18–0.44). From 1990 to 2019, the age-standardized DALY rate of VAD declined in all SDI regions. The lowest decrease occurred in low-SDI regions where the EAPC was greatest (−2.30%, 95% CI: −2.51% to −2.07%), whereas the greatest decrease occurred in the high-SDI group with the smallest EAPC (−3.49%, 95% CI: −4.03% to −2.92%) (Table 2, Figure 1B, Appendix A).

In addition, the EAPC in the age-standardized DALY rate of VAD was positively correlated with the age-standardized DALY rate (ρ = 0.376, *p* < 0.001) but negatively associated with the SDI (ρ = −0.327, *p* < 0.001, Appendix A). The proportion of DALY number was highest in people aged < 20 years old globally in all SDI regions, with proportions of almost 100% in some high-SDI regions. Low-SDI regions had the highest proportion of DALY number among children, and there was little change from 1990 to 2019. In contrast, the proportion of DALY number among children declined in other SDI regions (Appendix A). The annual proportions of DALY number decreased among people aged < 20 years but increased among adults aged > 20 years, whereas the least change was observed in low-SDI group (Appendix A). 

At the regional level, in 2019, the highest age-standardized DALY rate of VAD was observed in central sub-Saharan Africa (49.08 per 100,000; 95% UI: 32.59–69.88), followed by western sub-Saharan Africa (37.04 per 100,000; 95% UI: 25.35–51.76) and western sub-Saharan Africa (35.90 per 100,000; 95% UI: 24.49–50.85). The lowest age-standardized DALY rate was in Australasia (0.05 per 100,000; 95% UI: 0.03–0.09), followed by Eastern Europe (0.11 per 100,000; 95% UI: 0.06–0.18) and high-income North America (0.15 per 100,000; 95% UI: 0.08–0.27). From 1990 to 2019, the age-standardized DALY rate decreased least in Oceania (EAPC: −0.81%, 95% CI: −1.01% to −0.51%), followed by the Caribbean (EAPC: −0.89%, 95% CI: −0.98% to −0.82%) and central sub-Saharan Africa (EAPC: −1.07%, 95% CI: −1.48% to −0.51%) but decreased most in Eastern Europe (EAPC: −4.36%, 95% CI: −4.66% to −4.05%), followed by East Asia (EAPC: −4.07%, 95% CI: −4.44% to −3.78%) and Central Europe (EAPC: −3.83%, 95% CI: −3.97% to −3.73%) (Table 2, Figure 1B, and Appendix A).

In 2019, the countries with the highest age-standardized DALY rates of VAD were Niger (88.75 per 100,000; 95% UI: 57.91–128.32), Somalia (80.92 per 100,000; 95% UI: 52.54–118.45), and Mali (73.09 per 100,000; 95% UI: 46.88–105.53), while the countries with the lowest age-standardized DALY rates of VAD were Australia (0.01 per 100,000; 95% UI: 0.00–0.02), the Russian Federation (0.04 per 100,000; 95% UI: 0.02–0.08), and France (0.05 per 100,000; 95% UI: 0.02–0.11). From 1990 to 2019, the age-standardized DALY rates decreased in all countries except the Northern Mariana Islands (EAPC: 0.14%, 95% CI: −0.21% to 0.65%), Georgia (EAPC: 0.04%, 95% CI: −0.54% to 0.80%) and Kiribati (EAPC: −0.12%, 95% CI: −0.29% to 0.11%), where the age-standardized DALY rates remained stable. The countries with the greatest decrease in age-standardized DALY rates were Saudi Arabia (EAPC: −9.41%, 95% CI: −10.30% to −8.21%), Iran (Islamic Republic of) (EAPC: −8.73%, 95% CI: −9.25% to −8.32%), and the Maldives (EAPC: −7.82%, 95% CI: −8.47% to −7.29%) (Figure 2B, Appendix A). The global EAPCs in the age-standardized DALY rates of VAD in 204 countries and territories are shown in Appendix A, and the global age-standardized DALY rates of VAD in 204 countries and territories in 2019 are shown in Appendix A.

## 4. Discussion

We systematically analyzed temporal and spatial trends in the incidence and DALY rates of VAD at the global, regional, and national levels by sex, age, and SDI from 1990 to 2019. The global VAD incident number (−44.19%), age-standardized incidence rate (−59.85%), DALY number (−40.15%), and age-standardized DALY rate (−47.07%) declined from 1990 to 2019 due partly to continuous improvement in the global economy and the quality of health care as well as a wide range of food fortification and supplementation strategies. 

However, the burden of VAD remained high globally in 2019. The main causes of VAD include an insufficient intake of foods rich in vitamin A, the malabsorption of vitamin A, and a loss of vitamin A due to disease [7]. Although the trend of malnutrition has changed from undernourishment to excess energy intake in recent years, the consumption of micronutrient-rich foods remains far from sufficient [17]. Vitamin A supplementation (VAS) has been implemented in many regions and countries, but the complete implementation of periodic high-dose interventions is difficult in some countries with large populations; thus, the rate of coverage remains low [18]. Additionally, in some countries with poor sanitation, epidemics of infectious diseases have resulted in the loss of vitamin A [19] and thus increased the global burden of VAD.

The global burden of VAD differs hugely among regions with different SDIs, reflecting large discrepancies in medical facilities and health care. Low-SDI regions had high incidence and DALY rates at baseline and the slowest decline in age-standardized rates of VAD from 1990 to 2019, which caused the VAD burden to be much higher than that in regions with low-middle, middle, high-middle, and high SDI values in 2019; in fact, the age-standardized incidence rate was higher by almost a factor of 33, and the age-standardized DALY rate was higher by a factor of 131 in low-SDI regions than in high-SDI regions. Poverty in low-SDI regions is regarded as the main factor underlying the widespread occurrence of VAD [20]. The prevalence of VAD in low-income regions is high although the global mortality rate due to VAD has fallen sharply [21]. Another factor is low literacy [22]. Specifically, people with lower education levels may well suffer from VAD [23]. In regions with limited income and education, increasing the availability and utilization of vitamin A-rich foods through horticultural strategies might provide greater benefits than the large-scale adoption of VAS [24]. Implementing innovative social marketing strategies in addition to traditional nutrition education might be a more effective way to disseminate nutrition information [22].

At the regional level, in 2019, central, eastern, and western sub-Saharan Africa had the highest age-standardized incidence and DALY rates of VAD. These high rates are likely to be primarily due to the contributions of Somalia, Niger, and Mali, among other countries. In central sub-Saharan Africa in particular, the age-standardized rates of VAD declined slowly from 1990 to 2019. Falsified vitamin A (retinol) capsules with insufficient doses, severe degradation, and substandard specifications were widely circulated in Africa, the region with the most serious falsified medical products in the world [25]. It is imperative to crack down on the trafficking of counterfeit medicines. Orange-fleshed sweet potato, a variety of sweet potato rich in plant-based β-carotene, has been widely promoted in Africa for vitamin A fortification. However, the popularity and quality of orange-fleshed sweet potato remain issues that urgently need to be addressed [26]. It is necessary to increase the acceptance in the target population and improve the bioavailability of β-carotene by processing orange-fleshed sweet potato through local public sector programs and large retailers [27,28]. Additionally, research on other fortified staple foods, such as provitamin A carotenoid corn, needs to be promoted [29]. It is noted that properly preserving vitamin A drugs and food supplements, reducing food odors, increasing drug half-lives, and ensuring vitamin A activity are also important problems to be solved in sub-Saharan Africa, which has perennially high temperatures [30,31]. Failure to address these problems might also be related to widespread epidemics of malaria and acquired immunodeficiency syndrome in sub-Saharan Africa [32]. Furthermore, strengthening the coordination of interventions for overlapping epidemic diseases and VAD is an important direction in which to improve public health [19].

The WHO reported that the prevalence of VAD in Southeast Asia was as high as 50% [7]. We found that from 1990 to 2019, the age-standardized incidence and DALY rates of VAD in Southeast Asia decreased significantly, respectively ranking second and fourth among 21 regions, which reflects the great contribution of Southeast Asia in controlling the incident cases of VAD and thus reducing the global burden. Over the past 20 years, the increase times in incident cases and DALYs have been high, and the age-standardized rates have declined most slowly in Oceania, mainly owing to the situations in the Federated States of Micronesia, Kiribati, and the Northern Mariana Islands, which may be related to the dietary shift toward imported processed foods and lifestyle changes [33]. It is necessary to strengthen the awareness of VAD in these countries. Australia, as the most populous country in Oceania and having the lowest VAD burden worldwide, provides an important reference for these countries [34].

Children younger than 5 years of age, whose growth and development will affect the human capital and economic status of society in the future, are the main group susceptible to VAD [35]. Children with VAD are also more susceptible to respiratory infection, diarrhea, and parasitic infection. Repeated infections further reduce the absorption of vitamin A, resulting in a vicious cycle in children [36]. We found that the incidence and DALY rates of VAD declined with age globally, with the highest proportion of VAD cases being in children younger than 5 years of age in all SDI regions. The annual number of incident cases and DALYs of VAD in most parts of the world decreased in adolescents and children (<20 years of age) but increased in adults, which might be attributed to the global emphasis on the growth, development, and nutrition of adolescents and children as well as global aging. Furthermore, nutritional fortification might have a certain age effect [8].

The annual proportion of cases of VAD in low-SDI regions did not differ significantly between the age groups. With an increase in the SDI, the male-to-female ratio of the incidence of VAD was more biased toward the younger age group, whereas the male-to-female ratio of the DALY rate had the opposite bias. This might be because a greater awareness of health and availability of advanced diagnostic techniques in high-SDI regions resulted in a higher VAD detection rate in children and adolescents [37]. Meanwhile, increases in population aging and life expectancy bring increased risks of health problems related to VAD [38]. Regions with low SDI values contributed most to the global male-to-female ratio of the incidence and DALYs of VAD, indicating that the diagnosis of and intervention for VAD are imperative in regions with low SDI values, especially for children younger than 5 years of age. There might be additional children with undiagnosed or marginal VAD [39], which might have resulted in underestimation of the VAD burden in children. The use of VAS has been shown to significantly reduce VAD-related morbidity and mortality in children between 6 months and 5 years of age [18]. It is thus necessary to implement universal policies on VAS to prevent VAD in this age group.

Interestingly, the burden of VAD differed between males and females, with the age-standardized incidence and DALY rates being higher in males than in females. The male sex is considered a risk factor for VAD [40]. Additionally, the serum vitamin A concentration is negatively correlated with the occurrence of some male diseases, such as prostate cancer [41]. VAS has been implemented and combined with immunization programs in many regions and countries, and the interaction between VAS and vaccination is stronger in females [42]. This suggests that females have a greater awareness of micronutrient control, such as VAS and adjustment of the dietary structure, than males. Sex differences in the burden of VAD were more obvious in regions with low SDI levels. Male children might contribute to these figures more than female children because preferences for sons remains widespread in regions with low SDI values [43].

To the best of our information, this is the first report to describe the latest epidemiological patterns of VAD at the global, regional, and national levels and by sex, age, and SDI. The strength of this research is the systematic use of the latest GBD data from 2019 in assessing the incidence and DALY rates of VAD in various regions of the world from 1990 to 2019. The main limitation of this study is that owing to the potentially incomplete registration of VAD incident cases in some low-SDI regions, such as Africa and Latin America, during the GBD study period, the quality and accuracy of GBD data for such regions cannot be guaranteed, which might have affected the regional VAD burden analysis.

## 5. Conclusions

Despite the worldwide decrease in the age-standardized incidence and DALY rates of VAD from 1990 to 2019, the burden of VAD remains high in low-SDI regions, including most of sub-Saharan Africa and especially central sub-Saharan Africa. Globally, children younger than 5 years of age are most affected by VAD, and males are more susceptible than females to VAD. Therefore, attention should be paid to improving early prevention and treatment strategies for VAD in low-SDI regions, such as sub-Saharan Africa, especially in children younger than 5 years of age, to reduce the global burden of VAD.

## Figures and Tables

**Figure 1 nutrients-14-00950-f001:**
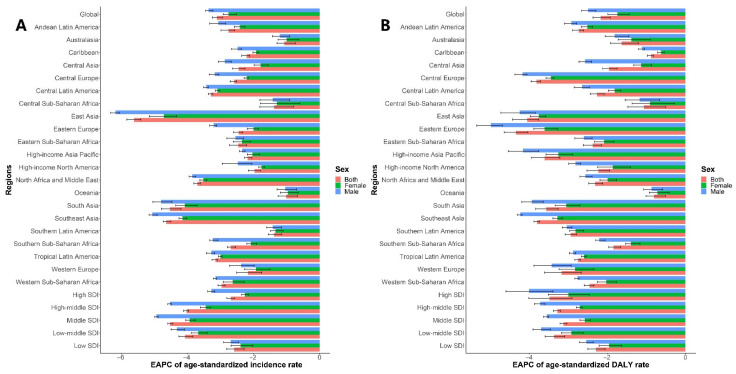
The EAPC of VAD age-standardized rates from 1990 to 2019 by sexes and regions. (**A**) The EAPC of age-standardized incidence rate. (**B**) The EAPC of age-standardized DALY rate. Abbreviations: DALY, disability adjusted life-year; EAPC, estimated annual percentage change; SDI, socio-demographic index; VAD, vitamin A deficiency.

**Figure 2 nutrients-14-00950-f002:**
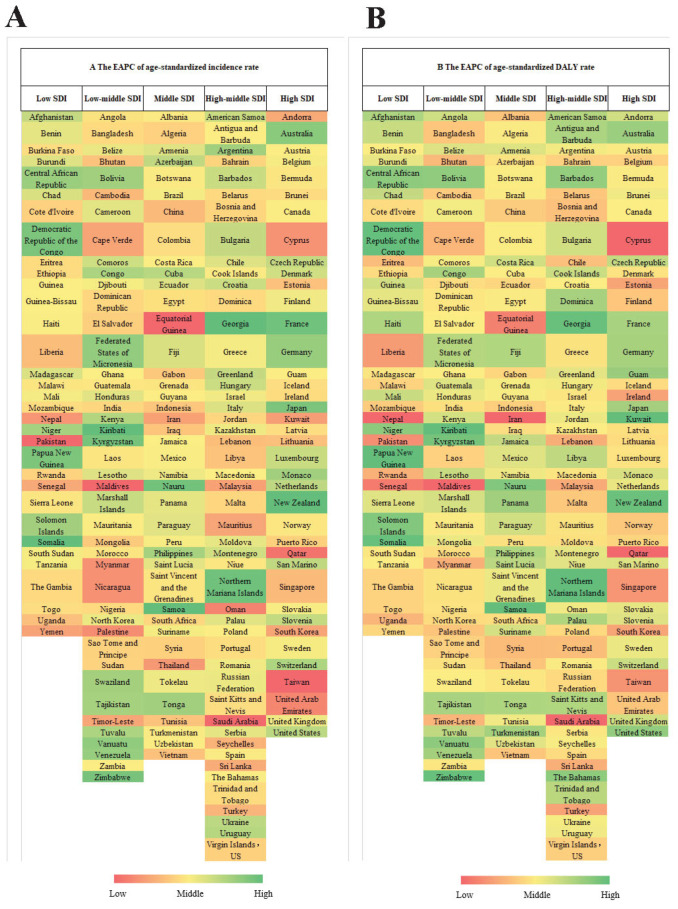
The EAPC of VAD age-standardized rates from 1990 to 2019 by countries. (**A**) The EAPC of age-standardized incidence rate. (**B**) The EAPC of age-standardized DALY rate. Abbreviations: DALY, disability adjusted life-year; EAPC, estimated annual percentage change; SDI, socio-demographic index; VAD, vitamin A deficiency.

**Table 1 nutrients-14-00950-t001:** The age-standardized incidence rate of VAD in 1990 and 2019 and its temporal trends.

	1990		2019		1990–2019	
	Age-Standardized Incidence Rate (per 100,000)		Age-Standardized Incidence Rate (per 100,000)		Change in Number No. (%)	EAPC
	No. (95% UI)	Male/Female Ratio	No. (95% UI)	Male/Female Ratio		No. (95% CI)
**Overall**	17,323.23 (16,526.51, 18,138.92)	1.57	6955.65 (6645.87, 7294.23)	1.31	−59.85%	−3.11 (−3.24, −2.94)
**Sex**						
Male	21,073.77 (19,711.46, 22,557.32)	NA	7886.24 (7367.67, 8489.75)	NA	−62.58%	−3.36 (−3.45, −3.23)
Female	13,456.51 (12,848.51, 14,141.38)	NA	5999.11 (5719.02, 6307.27)	NA	−55.42%	−2.75 (−2.92, −2.51)
**Sociodemographic index**						
Low	40,460.46 (39,202.05, 41,687.35)	1.57	19,156.10 (18,357.37, 19,950.88)	1.46	−52.65%	−2.58 (−2.81, −2.28)
Low-middle	29,005.62 (27,110.79, 30,890.15)	1.62	8877.03 (8202.49, 9642.84)	1.36	−69.40%	−4.07 (−4.26, −3.83)
Middle	14,790.69 (13,836.85, 15,869.70)	1.61	3805.06 (3559.91, 4080.13)	1.18	−74.27%	−4.53 (−4.59, −4.44)
High-middle	8176.35 (7576.08, 8905.80)	1.35	2575.67 (2431.78, 2737.05)	0.96	−68.50%	−4.02 (−4.11, −3.96)
High	1338.23 (1268.38, 1413.80)	0.80	586.69 (550.35, 625.85)	0.60	−56.16%	−2.69 (−2.80, −2.58)
**Region**						
Andean Latin America	12,877.69 (11,631.86, 14,260.37)	1.18	5904.69 (5259.36, 6633.17)	1.03	−54.15%	−2.75 (−2.98, −2.56)
Australasia	237.41 (210.38, 270.03)	0.59	148.85 (131.41, 168.77)	0.59	−37.30%	−1.07 (−1.29, −0.74)
Caribbean	11,146.70 (10,301.07, 11,991.62)	1.19	6289.74 (5659.18, 6996.38)	1.05	−43.57%	−2.21 (−2.35, −2.12)
Central Asia	8840.26 (8009.36, 9737.17)	1.96	4272.03 (3883.12, 4669.62)	1.48	−51.68%	−2.45 (−2.64, −2.25)
Central Europe	15,274.02 (14,477.09, 16,139.86)	0.71	7477.38 (7089.51, 7916.52)	0.57	−51.05%	−2.59 (−2.70, −2.51)
Central Latin America	14,629.63 (13,257.67, 16,046.77)	1.56	5717.55 (5096.11, 6419.49)	1.43	−60.92%	−3.29 (−3.37, −3.23)
Central Sub-Saharan Africa	43,280.37 (39,834.15, 46,536.67)	1.80	25,905.22 (23,288.44, 28,883.03)	1.79	−40.15%	−1.37 (−1.80, −0.79)
East Asia	11,210.77 (9382.33, 13,316.53)	1.99	2183.64 (1847.06, 2605.03)	1.18	−80.52%	−5.61 (−5.83, −5.41)
Eastern Europe	1071.28 (979.53, 1179.11)	0.77	530.88 (481.60, 587.02)	0.56	−50.44%	−2.45 (−2.61, −2.33)
Eastern Sub-Saharan Africa	46,770.60 (45,268.61, 48,255.91)	1.32	23,500.02 (22,337.24, 24,765.11)	1.27	−49.75%	−2.46 (−2.72, −2.21)
High-income Asia Pacific	1376.91 (1194.36, 1589.88)	0.82	683.87 (599.04, 779.97)	0.80	−50.33%	−2.18 (−2.27, −2.05)
High-income North America	811.05 (703.77, 924.89)	0.39	485.64 (408.06, 574.05)	0.36	−40.12%	−1.97 (−2.16, −1.79)
North Africa and Middle East	15,427.65 (14,749.24, 16,089.90)	1.37	5249.91 (4905.88, 5602.53)	1.26	−65.97%	−3.70 (−3.80, −3.60)
Oceania	19,889.72 (18,034.35, 21,935.52)	1.42	13,011.64 (11,381.72, 14,879.79)	1.36	−34.58%	−1.01 (−1.25, −0.67)
South Asia	27,177.40 (24,364.57, 30,028.67)	1.69	7189.30 (6181.29, 8388.98)	1.37	−73.55%	−4.53 (−4.79, −4.19)
Southeast Asia	21,792.80 (20,300.59, 23,360.26)	1.32	5175.32 (4667.95, 5733.85)	1.01	−76.25%	−4.64 (−4.73, −4.50)
Southern Latin America	10,805.24 (9611.36, 12,213.65)	1.44	6672.80 (5811.84, 7650.26)	1.40	−38.24%	−1.38 (−1.55, −1.15)
Southern Sub-Saharan Africa	18,153.70 (16,488.62, 20,003.67)	1.38	7834.62 (7001.01, 8690.17)	1.03	−56.84%	−2.69 (−2.79, −2.54)
Tropical Latin America	24,605.66 (22,010.20, 27,545.33)	1.27	10,005.39 (8592.70, 11,600.54)	1.21	−59.34%	−3.15 (−3.25, −3.08)
Western Europe	1408.59 (1320.28, 1508.58)	1.17	683.24 (637.29, 735.61)	1.08	−51.49%	−2.16 (−2.51, −1.75)
Western Sub-Saharan Africa	36,703.57 (35,417.46, 38,048.52)	1.93	15,570.91 (14,825.25, 16,315.94)	1.73	−57.58%	−2.97 (−3.07, −2.86)

Abbreviations: CI, confidence interval; EAPC, estimated annual percentage change; NA, not applicable; UI, uncertainty interval; VAD, vitamin A deficiency.

**Table 2 nutrients-14-00950-t002:** The age-standardized DALY rate of VAD in 1990 and 2019 and its temporal trends.

	1990		2019		1990–2019	
	Age-Standardized DALY Rate (per 100,000)		Age-Standardized DALY Rate (per 100,000)		Change in Number No. (%)	EAPC
	No. (95% UI)	Male/Female Ratio	No. (95% UI)	Male/Female Ratio		No. (95% CI)
**Overall**	31.95 (22.11, 45.30)	1.42	16.91 (11.53, 23.47)	1.17	−47.07%	−2.18 (−2.38, −1.93)
**Sex**						
Male	37.29 (25.60, 53.04)	NA	18.20 (12.41, 25.16)	NA	−51.20%	−2.50 (−2.67, −2.30)
Female	26.30 (17.83, 37.37)	NA	15.54 (10.55, 21.80)	NA	−40.91%	−1.75 (−1.99, −1.44)
**Sociodemographic index**						
Low	73.54 (50.66, 102.90)	1.44	38.02 (26.12, 52.78)	1.24	−48.29%	−2.30 (−2.51, −2.07)
Low-middle	50.95 (35.19, 72.27)	1.42	19.32 (12.93, 27.45)	1.16	−62.07%	−3.38 (−3.61, −3.11)
Middle	21.53 (14.48, 30.75)	1.37	8.61 (5.75, 12.42)	1.06	−60.03%	−3.14 (−3.22, −3.05)
High-middle	9.28 (6.17, 13.25)	1.40	3.74 (2.49, 5.43)	1.08	−59.68%	−3.29 (−3.39, −3.21)
High	0.91 (0.58, 1.35)	1.00	0.29 (0.18, 0.44)	0.77	−67.90%	−3.49 (−4.03, −2.92)
**Region**						
Andean Latin America	19.29 (12.72, 27.67)	1.23	9.14 (5.99, 13.34)	1.10	−52.61%	−2.74 (−2.90, −2.62)
Australasia	0.10 (0.05, 0.19)	1.31	0.05 (0.03, 0.09)	1.24	−50.11%	−1.64 (−1.92, −1.21)
Caribbean	16.71 (10.96, 24.06)	1.18	12.76 (8.29, 19.05)	1.05	−23.69%	−0.89 (−0.98, −0.82)
Central Asia	13.52 (8.72, 19.67)	1.54	7.59 (5.02, 11.02)	1.08	−43.84%	−1.96 (−2.13, −1.76)
Central Europe	10.14 (6.45, 15.08)	1.03	3.55 (2.22, 5.43)	0.86	−64.97%	−3.83 (−3.97, −3.73)
Central Latin America	14.99 (9.90, 21.27)	1.23	7.63 (5.06, 10.97)	0.94	−49.08%	−2.27 (−2.45, −2.09)
Central Sub-Saharan Africa	73.73 (48.76, 103.15)	1.42	49.08 (32.59, 69.88)	1.37	−33.43%	−1.07 (−1.48, −0.51)
East Asia	10.80 (6.95, 15.80)	1.70	3.75 (2.40, 5.64)	1.52	−65.26%	−4.07 (−4.44, −3.78)
Eastern Europe	0.35 (0.22, 0.55)	1.28	0.11 (0.06, 0.18)	0.92	−70.18%	−4.36 (−4.66, −4.05)
Eastern Sub-Saharan Africa	69.27 (47.83, 97.67)	1.27	35.90 (24.49, 50.85)	1.13	−48.18%	−2.38 (−2.63, −2.16)
High-income Asia Pacific	1.14 (0.64, 1.84)	0.70	0.35 (0.19, 0.60)	0.55	−69.02%	−3.62 (−3.96, −3.23)
High-income North America	0.29 (0.16, 0.49)	0.61	0.15 (0.08, 0.27)	0.55	−49.15%	−2.24 (−2.53, −1.95)
North Africa and Middle East	20.07 (13.53, 28.81)	1.32	10.17 (6.74, 14.32)	1.17	−49.35%	−2.32 (−2.48, −2.13)
Oceania	26.96 (17.73, 39.26)	1.30	18.30 (11.19, 27.04)	1.24	−32.11%	−0.81 (−1.01, −0.51)
South Asia	55.94 (38.43, 79.56)	1.44	20.24 (13.41, 29.10)	1.12	−63.81%	−3.58 (−3.85, −3.27)
Southeast Asia	38.43 (26.06, 56.15)	1.31	12.68 (8.35, 18.34)	1.03	−67.01%	−3.82 (−3.89, −3.75)
Southern Latin America	10.14 (6.05, 15.65)	1.44	4.24 (2.37, 7.09)	1.36	−58.18%	−2.95 (−3.09, −2.80)
Southern Sub-Saharan Africa	29.88 (19.56, 42.74)	1.34	16.42 (11.00, 23.58)	1.10	−45.07%	−1.85 (−1.97, −1.67)
Tropical Latin America	22.99 (14.52, 34.01)	1.06	10.17 (6.22, 15.05)	1.00	−55.77%	−2.76 (−2.84, −2.70)
Western Europe	0.82 (0.51, 1.26)	1.32	0.29 (0.17, 0.47)	1.15	−64.79%	−3.18 (−3.62, −2.68)
Western Sub-Saharan Africa	74.45 (50.41, 104.17)	1.64	37.04 (25.35, 51.76)	1.35	−50.24%	−2.47 (−2.60, −2.35)

Abbreviations: CI, confidence interval; DALY, disability adjusted life-year; EAPC, estimated annual percentage change; NA, not applicable; UI, uncertainty interval; VAD, vitamin A deficiency.

## Data Availability

Data are available in a public, open access repository. All of the data are publicly available. Data are available on request.

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
