# Peer review of "Global Burden of Vitamin A Deficiency in 204 Countries and Territories from 1990–2019"

_nutrients, 2022, doi:10.3390/nu14050950_

Round 1
Reviewer 1 Report
The research study is in an important area. Vitamin A deficiency is still a significant problem worldwide, particularly in the lower socioeconomic areas of the world as pointed out in the study.
A few comments should be addressed.
It would be helpful to have an abbreviation section in the article.
line 273 ...marketing strategies "instead of" traditional nutrition education. A better way to phrase it would be to substitute "in addition to" for "instead of." I do not think the authors want to replace traditional nutrition education.
With regard to the statement in lines 280-282, it could be that the lower vitamin A content of the vitamin A capsules could have been due to the storage of the vitamin A at high temperatures. The areas where there are problems with VAD, sub-Saharan Africa are hot and vitamin A is known to be destroyed by heat.
line 283 Was the sweet potato consumed with adequate fat necessary for beta-carotene and retinal (beta-carotene product) absorption.
Overall, a thorough and useful study.
Author Response
Response to Reviewer 1 Comments
Dear Reviewers,
We greatly appreciate the Reviewer’s comments and suggestions, which are very helpful in improving our manuscript. We have carefully revised the manuscript according to your comments and suggestions. Please kindly find the revised manuscript; point-by-point responses as given below.
Point 1:It would be helpful to have an abbreviation section in the article.
Response 1: Thank you very much for your careful review. We have added abbreviations in alphabetical order in lines 386-389 of the manuscript.
Point 2:line 273 ...marketing strategies "instead of" traditional nutrition education. A better way to phrase it would be to substitute "in addition to" for "instead of." I do not think the authors want to replace traditional nutrition education.
Response 2: We agree with your suggestion. "instead of" is too absolute. We have replaced it with "in addition to". Therefore, you can read " Implementing innovative social marketing strategies in addition to traditional nutrition education might be a more effective way to disseminate nutrition information " in lines 273-275 of the revised manuscript.
Point 3: With regard to the statement in lines 280-282, it could be that the lower vitamin A content of the vitamin A capsules could have been due to the storage of the vitamin A at high temperatures. The areas where there are problems with VAD, sub-Saharan Africa are hot and vitamin A is known to be destroyed by heat.
Response 3: Thanks for this interesting question. As you suggested, temperature plays an important role in the chemical stability and storage conditions of vitamin A. We found that, in fact, there are similar problems not only in drugs, but also in other forms of vitamin A supplementation (such as biofortified staple foods rich in carotenoids) in Africa, resulting in partial or total loss of vitamin A activity. Degradation of vitamin A can produce odor, and reduce consumer acceptance and nutritional intake [1]. In view of this phenomenon, there are mainly the following solutions. For foods, changing the post-harvest processing method, such as changing the traits, controlling storage temperature [2], or using the packaging material of double-layered polyethylene can improve the stability of carotenoids [1]. For drugs, using stabilizers such as butylated hydroxytoluene or hydroxyanisole to configure retinol, changing the emulsification system, adding nanoparticles, or freeze-drying increase the stability of retinol and the half-life of drugs [3, 4].
However, it is reported that the total cost of vitamin A capsules in Africa is currently the lowest in the world [5], and it is difficult for this region to invest more money in the storage of vitamin A capsules. Therefore, you can read "It is noted that properly preserving vitamin A drugs and food supplements, reducing food odors, increasing drug half-lives, and ensuring vitamin A activity are also important problems to be solved in sub-Saharan Africa, which has perennially high temperatures" in lines 290-293 of the revised manuscript.
[1] Ekpa O, Fogliano V, Linnemann A. Carotenoid stability and aroma retention during the post-harvest storage of biofortified maize. J Sci Food Agric 2021 Aug; 101(10):4042-4049. Epub 2021 Jan 8. doi: 10.1002/jsfa.11039.
[2] Barrett AH, Richardson MJ, Froio DF, Connor LFO, Anderson DJ, Ndou TV. Long-Term Vitamin Stabilization in Low Moisture Products for NASA: Techniques and Three-Year Vitamin Retention, Sensory, and Texture Results. J Food Sci 2018 Aug; 83(8):2183-2190. Epub 2018 Jul 30. doi: 10.1111/1750-3841.14218.
[3] Eskandar NG, Simovic S, Prestidge CA. Chemical stability and phase distribution of all-trans-retinol in nanoparticle-coated emulsions. Int J Pharm 2009 Jul; 376(1-2):186-94. Epub 2009 May 5. doi: 10.1016/j.ijpharm.2009.04.036.
[4] Ghouchi-Eskandar N, Simovic S, Prestidge CA. Solid-state nanoparticle coated emulsions for encapsulation and improving the chemical stability of all-trans-retinol. Int J Pharm 2012 Feb; 423(2):384-391. Epub 2011 Dec 23. doi: 10.1016/j.ijpharm.2011.12.027.
[5] Neidecker-Gonzales O, Nestel P, Bouis H. Estimating the global costs of vitamin A capsule supplementation: a review of the literature. Food Nutr Bull 2007; 28(3):307-316. doi: 10.1177/156482650702800307.
Point 4: line 283 Was the sweet potato consumed with adequate fat necessary for beta-carotene and retinal (beta-carotene product) absorption.
Response 4: Thanks for your question to our discussion. This question has also aroused our interest. We reviewed the relevant literature and explained this issue as follows. Sweet potato can be divided into five varieties according to the color of flesh: white, cream, yellow, orange, and purple. Orange-fleshed sweet potato (OFSP) is the only one that contains high concentration of provitamin A carotenoids, especially β- Carotene, a variety that the body can convert into vitamin A [1]. However, as you mentioned, a systematically observe to digestion, absorption, and metabolism is required when considering the solution of nutrient deficiency. Current research on the bioavailability of β-carotene in OFSP is limited, but dietary intervention studies have shown that the vitamin A status of school-age children after eating OFSP is significantly improved [2,3].
The bioavailability of β-carotene depends on the presence of dietary fat, which is not provided by OFSP. Some researchers have found that adding fat source to OFSP can improve the bioavailability of β- carotene and the subsequent vitamin A status [4,5], and adding a certain amount of peanut paste (rich in monounsaturated lipid, which has been proved to promote the absorption of carotenoids [6]), to develop a β- carotenoid-enriched food products by computer modeling [7]. In fact, after the formation of vitamin A, the execution of physiological functions requires sufficient protein, which is also lacking in OFSP. Therefore, in addition to fat, the researchers also have added chickpea to supplement some limiting amino acids [7]. In view of the current promotion status of OFSP in sub-Saharan Africa, product quality is the main factor that should be considered, and processing of OFSP is an opportunity to be developed [8].
Therefore, you can read, " Orange-fleshed sweet potato, a variety of sweet potato rich in plant-based β-carotene, has been widely promoted in Africa for vitamin A fortification. However, the popularity and quality of orange-fleshed sweet potato remain issues that urgently need to be addressed [26]. It is necessary to increase the acceptance in the target population, and improve the bioavailability of β-carotene by processing orange-fleshed sweet potato through local pub-lic sector programs and large retailers [27-28]. Additionally, research on other fortified staple foods, such as provitamin A carotenoid corn, needs to be promoted." in lines 283-290 of the revised manuscript.
[1] Low JW, Mwanga ROM, Andrade M, Carey E, Ball AM. Tackling vitamin A deficiency with biofortified sweetpotato in sub-Saharan Africa. Glob Food Sec 2017; 14: 23-30. doi: 10.1016/j.gfs.2017.01.004.
[2] van Jaarsveld PJ, Faber M, Tanumihardjo SA, Nestel P, Lombard CJ, Benadé AJ. Beta-carotene-rich orange-fleshed sweet potato improves the vitamin A status of primary school children assessed with the modified-relative-dose-response test. Am J Clin Nutr 2005; 81(5):1080-1087. doi: 10.1093/ajcn/81.5.1080.
[3] Low JW, Arimond M, Osman N, Cunguara B, Zano F, Tschirley D. A food-based approach introducing orange-fleshed sweet potatoes increased vitamin A intake and serum retinol concentrations in young children in rural Mozambique. J Nutr 2007; 137(5):1320-1327. doi: 10.1093/jn/137.5.1320.
[4] Muzhingi T, Yeum KJ, Bermudez O, Tang G, Siwela AH. Peanut butter increases the bioavailability and bioconversion of kale β-carotene to vitamin A. Asia Pac J Clin Nutr 2017; 26(6):1039-1047. doi: 10.6133/apjcn.112016.03.
[5] Bengtsson A, Larsson Alminger M, Svanberg U. In vitro bioaccessibility of beta-carotene from heat-processed orange-fleshed sweet potato. J Agric Food Chem 2009; 57(20):9693-9698. doi: 10.1021/jf901692r.
[6] Goltz SR, Campbell WW, Chitchumroonchokchai C, Failla ML, Ferruzzi MG. Meal triacylglycerol profile modulates postprandial absorption of carotenoids in humans. Mol Nutr Food Res 2012; 56(6):866-877. doi: 10.1002/mnfr.201100687.
[7] Lewandowski K, Zhang X, Hayes M, Ferruzzi MG, Paton CM. Design and Nutrient Analysis of a Carotenoid-Rich Food Product to Address Vitamin A and Protein Deficiency. Foods 2021; 10(5):1019. doi: 10.3390/foods10051019.
[8] Laurie SM, Faber M, Claasen N. Incorporating orange-fleshed sweet potato into the food system as a strategy for improved nutrition: The context of South Africa. Food Res Int 2018 Feb; 104:77-85. Epub 2017 Sep 9. doi: 10.1016/j.foodres.2017.09.016.
Finally, we have revised and improved the language of the full text. The modified location is marked in red. The attachment is the revised manuscript.
Thank you again for your rigorous approach to our work.

Reviewer 2 Report
In this paper, Tian Zhao et al. aim to estimate the incidence and DALYs of VAD at global, regional and national levels using data from the 2019 Global Burden of Disease Study.
Scientific level of the paper is good and statistical approach is fine. However, the article could be improved by carrying out editing of the Discussion section by a native speaker of English.
For example :
line 266 : replace "were" with "to be". Furthermore, please specify "with other" SDI.
line 276 : "in these regions" : it is not needed.
Author Response
Response to Reviewer 2 Comments
Dear Reviewers,
We greatly appreciate the Reviewer’s comments and suggestions, which are very helpful in improving our manuscript. We have carefully revised the manuscript according to your comments and suggestions. Please kindly find the revised manuscript; point-by-point responses as given below.
Point 1: line 266: replace "were" with "to be". Furthermore, please specify "with other" SDI.
Response 1: We agree with your suggestion. We have replaced "were" with "to be", and replaced "with other" with "with low-middle, middle, high-middle, and high". Therefore, you can read "which caused the VAD burden to be much higher than that in regions with low-middle, middle, high-middle, and high SDI values in 2019" in lines 266-267 of the revised manuscript.
Point 2: line 276: "in these regions" : it is not needed.
Response 2: As suggested, we have deleted "in these regions", you can read "At the regional level, in 2019, Central, Eastern, and Western sub-Saharan Africa had the highest age-standardized incidence and DALY rates of VAD" in lines 276-277 of the revised manuscript.
Finally, we have revised and improved the language of the full text. The modified location is marked in red. The attachment is the revised manuscript.
Thank you again for your rigorous approach to our work.
